# circRNAome Profiling in Oral Carcinoma Unveils a Novel circFLNB that Mediates Tumour Growth-Regulating Transcriptional Response

**DOI:** 10.3390/cells9081868

**Published:** 2020-08-10

**Authors:** Yi-Tung Chen, Ian Yi-Feng Chang, Chia-Hua Kan, Yu-Hao Liu, Yu-Ping Kuo, Hsin-Hao Tseng, Hsing-Chun Chen, Hsuan Liu, Yu-Sun Chang, Jau-Song Yu, Kai-Ping Chang, Bertrand Chin-Ming Tan

**Affiliations:** 1Department of Biomedical Sciences, College of Medicine, Chang Gung University, Taoyuan 333, Taiwan; love5p3@yahoo.com.tw (Y.-T.C.); a604256008@gmail.com (C.-H.K.); erinchen25@gmail.com (H.-C.C.); 2Research Center for Emerging Viral Infections, Chang Gung University, Taoyuan 333, Taiwan; 3Molecular Medicine Research Center, Chang Gung University, Taoyuan 333, Taiwan; ianyfchang@mail.cgu.edu.tw (I.Y.-F.C.); ellen0918@gmail.com (Y.-P.K.); liu-hsuan@mail.cgu.edu.tw (H.L.); ysc@mail.cgu.edu.tw (Y.-S.C.); yusong@mail.cgu.edu.tw (J.-S.Y.); dr.kpchang@gmail.com (K.-P.C.); 4Graduate Institute of Biomedical Sciences, College of Medicine, Chang Gung University, Taoyuan 333, Taiwan; ericliu0118@gmail.com (Y.-H.L.); ggasdqwe@gmail.com (H.-H.T.); 5Department of Cell and Molecular Biology, College of Medicine, Chang Gung University, Taoyuan 333, Taiwan; 6Division of Colon and Rectal Surgery, Lin-Kou Medical Center, Chang Gung Memorial Hospital, Taoyuan 333, Taiwan; 7Department of Otolaryngology-Head & Neck Surgery, Lin-Kou Medical Center, Chang Gung Memorial Hospital, Taoyuan 333, Taiwan; 8Department of Neurosurgery, Lin-Kou Medical Center, Chang Gung Memorial Hospital, Taoyuan 333, Taiwan

**Keywords:** oral cancer, non-coding RNA, circular RNA, miRNA–mRNA network, biomarkers

## Abstract

Deep sequencing technologies have revealed the once uncharted non-coding transcriptome of circular RNAs (circRNAs). Despite the lack of protein-coding potential, these unorthodox yet highly stable RNA species are known to act as critical gene regulatory hubs, particularly in malignancies. However, their mechanistic implications in tumor outcome and translational potential have not been fully resolved. Using RNA-seq data, we profiled the circRNAomes of tumor specimens derived from oral squamous cell carcinoma (OSCC), which is a prevalently diagnosed cancer with a persistently low survival rate. We further catalogued dysregulated circRNAs in connection with tumorigenic progression. Using comprehensive bioinformatics analyses focused on co-expression maps and miRNA-interaction networks, we delineated the regulatory networks that are centered on circRNAs. Interestingly, we identified a tumor-associated, pro-tumorigenic circRNA, named circFLNB, that was implicated in maintaining several tumor-associated phenotypes in vitro and in vivo. Correspondingly, transcriptome profiling of circFLNB-knockdown cells showed alterations in tumor-related genes. Integrated in silico analyses further deciphered the circFLNB-targeted gene network. Together, our current study demarcates the OSCC-associated circRNAome, and unveils a novel circRNA circuit with functional implication in OSCC progression. These systems-based findings broaden mechanistic understanding of oral malignancies and raise new prospects for translational medicine.

## 1. Introduction

With approximately 250,000 annual cases reported worldwide, oral cancer is a highly prevalent cancer and a growing health concern [1,2]. Oral squamous cell carcinoma (OSCC) is the most common subtype of oral cancer, and accounts for more than 90% of all cases. In addition to the known risk behaviors of cigarette smoking, alcohol use, and betel nut chewing, human papilloma virus (HPV) infection is another risk factor that has been found among OSCC patients [3,4,5]. Despite progress in diagnostic and therapeutic strategies in the last three decades, the five-year survival rate in patients with OSCC has not significantly improved due to aggressive local invasion, distant metastasis, and recurrence [6]. Thus, a better mechanistic understanding of OSCC progression may shed new light on this malignancy and possibly reveal new avenues for therapeutic interventions. Cumulative research has implicated multiple dysregulated molecular networks in OSCC neoplastic development, including the epidermal growth factor receptor (EGFR), Ras, NF-κB, STAT, Wnt/β-catenin, TGF-β, and PI3K-AKT-mTOR signaling pathways [7]. While intensive investigations over the past decades have focused on the role of protein-coding oncogenes or tumor-suppressor genes in the pathogenesis of OSCC, the contribution of alterations in the non-coding transcriptome to its incidence or progression has not yet been fully explored.

In recent years, deep sequencing technologies have aided the delineation of the non-coding constituents of the transcriptome. Despite the lack of protein-coding potential, these once uncharted transcriptomes have emerged as important determinants in gene regulation by acting as critical switches that fine-tune transcriptional and signaling output. Among the endogenous noncoding RNAs (ncRNAs), circular RNAs (circRNAs) represent an intriguing subgroup. The circRNAs are characterized by their unique biogenesis, which involve a back-splicing reaction that results in a covalently closed loop without a 5′-cap or a 3′-poly (A) tail [8]. Importantly, based on their distinct tissue-specific expression patterns and structural stability [9,10], they may be exploited as new tumor markers and/or treatment targets. Towards this end, functional implications of circRNAs in the process of cancer progression have recently been substantiated [11,12,13]. In particular, oral cancer-associated circRNAs have been identified and further demonstrated to be key regulators of the tumorigenic process [14,15,16]. At the mechanistic level, a predominant mode of circRNA-mediated regulation involves serving as competing endogenous RNA (ceRNA) [17], in which sequence complementarity contributes to the binding or “sponging” of their targeting miRNAs and leads to the consequent de-repression of the mRNA transcripts with shared cognate miRNA-targeted sequences [18,19].

Despite growing evidence for the tumorigenic role of circRNAs and their distinct regulatory action, research of these enigmatic molecules is still in its infancy, and only a limited number of cancer-related circRNAs have been comprehensively studied. Therefore, the systematic identification of tumor-associated circRNAs and a thorough understanding of their mechanisms are necessary for the realization of any translation medicine potential. To this end, we previously reported a deep sequencing-based identification of mutational signatures underlying advanced OSCC [20], in which alterations in the tumor-associated transcriptome were also profiled. Using this data set, we uncovered differentially expressed circRNAs, most of which were hitherto uncharacterized. Based on co-expression maps and miRNA-targeted interactions, we comprehensively constructed the circRNA-guided regulatory networks underlying OSCC. The expression and tumor-biased patterns of one circRNA, circFLNB, was further confirmed by structure-specific and quantitative PCR analyses of additional patient samples and oral cancer cell lines. The mature form of circFLNB is comprised of three exons and is intronless and predominantly distributed in the cytoplasm. Functionally, the silencing of circFLNB expression reduced cell proliferation and colony formation, but enhanced doxorubicin-mediated apoptosis in oral cancer cell lines. Additionally, downregulation of circFLNB abrogated the migratory and invasive abilities of cancer cells, further highlighting its significance as a tumor promoter. At the mechanistic level, we used RNA-seq and in silico analyses to demonstrate that circFLNB might regulate a set of tumor-related genes. In summary, our current study systematically demarcates distinct and hitherto unreported circRNAome signatures in association with OSCC and demonstrates their oncogenic potential by functionally characterizing a novel tumorigenic circRNA.

## 2. Materials and Methods

### 2.1. Sequencing Data Analysis and Bioinformatics Analysis

The processed RNA-seq data are transferred to Partek Genomics Suite and mapped for ncRNA or mRNA annotations on the basis of GENCODE v25. Partek Genomics Suite and statistical packages are used to perform statistical analysis, hierarchical clustering, ncRNA identification, differential expression analysis, and pathway enrichment. The ncRNA data will also be integrated with the TCGA dataset to pinpoint specific ncRNA candidates/signatures, which will be further selected for PCR-based verification.

### 2.2. RNA-Sequencing and Gene Expression Analysis

TRIzol reagent (Invitrogen, CA, USA) was used for RNA extraction, and cDNA libraries were prepared based on the TruSeq^®^ Stranded Total RNA Sample Preparation Guide (Illumina, Part # 15031048, San Diego, CA, USA). Equal concentrations of each library were sequenced using a NextSeq 500 (Illumina) platform to create pair-end 75-bp reads. Quality assessment and trimming of the generated sequences were done by the RNA-seq alignment tool from BaseSpace (Illumina), followed by alignment to the human reference genome (hg38) with STAR 2.5.2b [21]. The expression levels of genes in each sample and the corresponding fold changes were estimated by DESeq2 1.14.1 (Partek Genomics Suite, St. Louis, MO, USA) [22] with GENCODE V19 (V25) annotation [23]. Relative expression of each gene is represented by CPM (Counts Per kilobase Million). Partek Genomics Suite and statistical package were used for the statistical analysis, hierarchical clustering differential expression analysis, and Gene ontology (GO) enrichment. Canonical pathway analysis was performed by using the Ingenuity Pathway Analysis (IPA).

### 2.3. Circular RNA Identification and miRNA Sponge Prediction

We used Trimmomatics (Version 0.36) to remove the sequencing adapters and low-quality bases from RNA-seq FASTQ files. To detect circular RNAs by KNIFE (v1.4) [24], we first used the createJunctionIndex program of KNIFE to create a junction sequence database on the basis of human genome hg38 and GENCODE V25 annotations [23]. Next, we applied the findcircularrna.sh scripts with default parameters to the cleaned FASTQ files. We only considered the results reported by the GLM mode. All junctions with a posterior probability of 0.9 or higher were retrieved for subsequent differential expression analysis. To examine the potential miRNA sponge activity, we performed a two-part analysis: TargetScan 7.0 [25] was first employed to comprehensively identify putative miRNA binding sites in circRNAs and mRNAs. Given a presumably inverse correlation in expression between miRNA and mRNAs [26], we also incorporated co-expression analysis to determine circRNA–mRNA pairs with coordinated expression. Any co-expressed circRNA–mRNA pairs also harboring common miRNA target sites would be an indication that the corresponding circRNA candidates act as a miRNA sponge.

### 2.4. RNA Extraction, Reverse Transcription (RT)-PCR, and Quantitative PCR (qPCR)

Total RNA was extracted by TRIzol reagent, and reverse transcribed into complementary DNA (cDNA) by MML-V reverse transcriptase (Invitrogen) with random hexamer. Individual gene expression was analyzed by real-time quantitative PCR (iQ5 Gradient Real Time SYBR-Green PCR system) with specific primers and analyzed by CFX Manager Software (Bio-Rad, CA, USA). The relative gene expression was determined by normalization to the internal control gene expression, and the control group was represented as 1. All results were obtained from at least three independent experiments, and statistical significance was measured by the Student’s *t* test and presented in *p*-value form. All used primers were listed in Appendix A.

### 2.5. Cell Culture

SAS and HeLa cells were cultured in high-glucose Dulbecco’s modified Eagle’s medium, and 1× Non-Essential Amino Acid (NEAA) and 1 mM sodium pyruvate was added to the medium for OECM1 cells. SCC25 cells were cultured in Dulbecco’s Modified Eagle Medium: Nutrient Mixture F-12 containing 1× NEAA, 1 mM sodium pyruvate, and 400 ng/mL hydrocortisone. All culture media were supplemented with 10% heat-inactivated fetal bovine serum and 1 U/mL penicillin-streptomycin. All media and reagents were purchased from Thermo Fisher Scientific (Waltham, MA, USA). These cells were incubated at 37 °C with 5% CO_2_ in a humidified incubator.

### 2.6. MTT Cell Proliferation Assay and Colony Formation Assay

3-(4,5-Dimethylthiazol-2-yl)-2,5-diphenyltetrazolium bromide (MTT) cell proliferation assay was performed based on living cells containing mitochondrial dehydrogenase activity to metabolize MTT substrate. Briefly, 2.5 × 10^4^ cells were seeded in a 24-well culture plate and incubated with MTT reagent (Sigma, St. Louis, MO, USA) for 1 h, and formed precipitates were dissolved and quantified by spectrophotometry at 570 nm for determining cell viability. For the colony formation assay, 2.5 × 10^3^ cells were seeded in a 6-well plate for a 7-day culture, and forming colonies were stained by crystal violet. Cell clonogenicity was assessed by quantifying the colony area by ImageJ software (NIH, Bethesda, MD, USA).

### 2.7. Plasmids Construction for Gene Knockdown and Overexpression

RNAi-mediated gene silencing was performed by using the pLKO-TRC017 RNAi system, with the target sequences annealed and ligated into the TRC017 vectors. For constructing the circFLNB expression vector, the exonic region of circFLNB RNA and the flanking Alu elements were amplified from the cDNA sample. The resulting PCR products were ligated into the cloning vector using the HE Swift Cloning Kit (TOOLS Life Science, Taiwan), and subsequently sub-cloned into the expression vector pcDNA3.1 (−) and lentiviral-specific pLAS2W vector. All primer sequences are listed in Appendix A. Constructed plasmids were packaged into viral particles with 293FT cells, and infected into oral cancer cell lines to alter specific gene expression. All experimental procedures were conducted based on manufacturer’s instructions (RNAi core, Academia Sinica, Taiwan).

### 2.8. Wound Healing Assay and Transwell Experiments

For wound healing assay, 1.2 × 10^6^ cancer cells were seeded in six-well plate, and which were subsequently scratched by a 20 μL pipette tip. Wounded cell migration was recorded via the Cytation^TM^ 5 Cell Imaging instrument (BioTek Instruments, Inc., Winooski, VT, USA). For Transwell migration and invasion assays, Transwell polystyrene membrane Insert (Corning, NY, USA) and Matrigel (BD Biosciences, San Jose, CA, USA) were used. Briefly, 10^5^ cells were seeded into the Transwell chamber coated with Matrigel (invasion) or without (for migration). Serum-free medium was added to the top of chambers, and the lower level was filled with culture medium. After 16 h incubation, the migrating (invading) cells through chambers were fixed and stained by crystal violet and counted by microscope.

### 2.9. In Vivo Mouse Xenograft Experiment

Male NOD.CB17-Prkdc^scid^/JNarl mice (6 weeks old) were provided by the National Laboratory Animal Center (NLAC). Specific cell lines were injected (1 × 10^6^) subcutaneously into rear flank of mice, using a 26-gauge needle. Tumor formation and growth curves were monitored by a Vernier caliper, and tumor volumes were calculated with the formula: length × width^2^ × 0.52. Tumor weight was measured after mice sacrifice. The animal experiment was approved by Laboratory Animal Center, Chang Gung University.

### 2.10. Ethics Approval and Consent to Participate

This study was approved by the Chang Gung Memorial Hospital Institutional Review Board as a retrospective analysis (201700774B0C501) and was conducted within the guidelines of the Declaration of Helsinki. Patients/families were counseled in the context of the present study design, and all participants provided written informed consent to participate in the study.

## 3. Results

### 3.1. Integrated Transcriptome-Wide Profiling of the OSCC-Associated circRNA Landscape

In our previous report on the mutation-based prognostic gene signature associated with advanced OSCC [20], we recruited ethnic Taiwanese patients admitted to the Chang Gung Memorial Hospital (*n* = 39) and completed RNA-seq for matched pairs of tumors and adjacent normal tissues from the same patients, thus totaling 78 datasets. The clinical characteristics and demographics of our cohort, as well as the detailed statistical analyses of sequencing data are presented therein. To comprehensively catalog the dysregulated circRNAome alterations underlying OSCC, we further processed the sequencing data into qualitatively and quantitatively profiled circRNAs, based on the expression of back-splicing junctions. For this purpose, an open-sourced tool, KNIFE [24], was employed to call back-splicing events from our RNA-seq data, which resulted in the identification of 113,972 species of circular RNAs.

To comparatively illustrate the overall circRNA transcriptome profiles among the specimens, principal component analysis (PCA) of the RNA-seq data was performed, consequently revealing distinct expression profiles corresponding to the disease states (Figure 1A). Next, circRNA genes exhibiting tumor-associated differential expression patterns were identified using Partek GS. A total of 443 (207 upregulated and 236 downregulated) circRNA species, derived from 382 parental coding genes, were found to be differentially represented in the OSCC tumor vs. normal tissues (|fold change| ≥ 2, *p*-value < 0.05, FDR < 0.001, Appendix A). The overall distributions of these circRNAome changes in relation to various clinical attributes were further characterized by hierarchical clustering and are shown as a heatmap (Figure 1B). Similar to the PCA plot, distinct clustering for normal and tumor tissues based on the circRNA profiles was evident, which strongly implied a correlation between dysregulated circRNA expression and the tumorigenesis process.

We further performed in silico characterization of the differentially expressed circRNAs (DECs) and made the following lines of observations. First, regarding transcript structure, the identified circRNAs were mostly classified as multiple-exonic type (89.6%) (Figure 1C, upper right panel). Second, the chromosomal origins of the OSCC-associated circRNA expression showed a rather stereotypical distribution for the back-splicing events, which corresponded with chromosome size (Figure 1C, lower panel). Third, circRNA abundance was largely correlated with their parental coding gene expression (Figure 2A,B). Finally, given that tumorigenic progression is typically attributed to alterations in molecular pathways, we also explored dysregulated pathways represented by our circRNA-encoding parental gene set. To this end, GO enrichment analysis revealed significant enrichment in several biological pathways, revealing the broad regulatory network by circRNA molecules (Figure 2C). These findings further hinted at the tumorigenic relevance of circRNA perturbations, which constitute an additional layer of gene networks in OSCC.

Concurrent profiling of expressed mRNAs and circRNAs constitutes a strong basis for the in-depth assessment of the regulatory relationships between these two types of RNAs in OSCC transcriptomes. To construct a global mRNA–circRNA interaction network, we compiled a co-expression map based on the 39 patient transcriptome data, in which 443 differentiated expressed circRNAs and 30,011 mRNAs with coordinated expression patterns (*p* < 0.05) were interconnected to form 3,108,927 unique circRNA–mRNA pairs. Further, owing to the widely reported miRNA-sponging activity of circRNAs, we expanded the regulatory hierarchies by incorporating miRNA-target interactions. Toward this end, we first retrieved computational miRNA-target interactions based on TargetScan predictions (Release 7.0) [27], and retained miRNAs with at least two potential targeting sites in any circRNA, or at least one site in any given mRNA 3′ UTR. A two-layer miRNA sponging axis was then established for positively correlated circRNA–mRNA pairs that were found to harbor shared miRNA targeting sites. This cross-referencing of sequencing data and informatics prediction captured an extensive transcriptome regulatory network putatively associated with OSCC tumorigenesis (Figure 2D), in which 319 miRNAs and 10,887 mRNAs were further co-aggregated into three-tier regulation sub-networks (*n* = 473,294). These analyses underscored the broad implications of circRNAs in cancer-associated transcriptome alterations and provided a mechanistic basis for their molecular regulation.

### 3.2. Identification and Validation of circRNAs Differentially Expressed in OSCC Patients

We implemented transcript abundance and statistical testing filtering in differential expression profiling on our extensive in-house database, and subsequently identified a set of previously uncharacterized circRNAs. Due to the uncertain nature of these distinctively structured RNAs, we performed PCR and Sanger sequencing experiments to independently verify their existence and their tumor-associated expression patterns. We first performed end-point PCR assays using specific divergent primers and subsequently demonstrated the differential expression profiles of circRNAs between the paired OSCC normal and tumor tissues. We found that circSNX5 and circFLNB expressions were elevated in tumor specimens, whereas circABCA6 and circBNC2 were downregulated (Figure 3A), which corroborated findings from the RNA-seq data (Appendix A). Because the structural complexity of transcripts may contribute to false-positive junction formation in the complementary DNA (cDNA) synthesis reaction, circRNAs can sometimes be erroneously identified as artificial byproducts of reverse transcription. To address this issue, we synthesized cDNAs with another commercially available transcriptase, SuperScript III, and repeated the PCR detection of the circRNA expression. Target circRNA expression was observed in templates synthesized by both MML-V and SuperScript transcriptases, thus affirming the presence of endogenous circRNA in patient samples (Figure 3B). Moreover, PCR amplicons were ligated into the pHE cloning vector for complete Sanger sequencing, which demonstrated the “head-to-tail” splice junction of circRNA as another means of experimental validation (Figure 3C and Appendix A). These experiments evidenced the expression of circRNAs and their differential expression in normal and OSCC tissues.

### 3.3. Characterization of an OSCC-Associated circRNA, circFLNB

Among the candidate circRNAs putatively associated with OSCC, we selected circFLNB for further functional characterization based on its prominent and consistent upregulation in tumor specimens in comparison with other candidates (Figure 3A). In an independent set of OSCC clinical specimens, we were able to verify the enhanced expression of circFLNB in tumor sections (Figure 3D), consistent with the above end-point PCR and high-throughput data. Additionally, junction-specific RT-PCR and quantitative PCR experiments demonstrated the presence of circFLNB RNA in oral cancer cell lines (Appendix A). In contrast, divergent amplification of the corresponding genomic DNA sample did not yield any product, which confirmed the specific detection for circFLNB RNA expression in this assay (Figure 3E and Appendix A). Next, to test the notion that the structural circularity of circRNA confers a greater resistance to exonuclease-mediated degradation, we monitored the turnover rate of circFLNB and the parental FLNB mRNA. While the abundance of the linear transcript counterpart dropped by nearly half after a 6 h treatment with Actinomycin D (Figure 3F), the expression of circFLNB remained virtually unchanged. The high stability of circFLNB may contribute to its functional role in oral cancer cells.

To further probe the post-transcriptional attributes of circFLNB, we then focused on its exon-intron organization and subcellular localization. Specific primers located on the circularized exons or adjoining introns were designed and used to pinpoint the sequence arrangement of the circFLNB molecule (Appendix A, upper panel). Gel electrophoresis of RT-PCR amplicons showed that introns were absent from the transcript (Appendix A, lower panel), indicating that circFLNB is primarily an exonic circRNA arising from FLNB exons 2 to 4. Next, a subcellular fractionation experiment was performed, in which U48 and 7SL RNA expression were measured to serve as markers for the nuclear and cytosolic compartments, respectively (Appendix A, lane 2 and 3). The cytoplasm-to-nuclei biased proportion of circFLNB expression indicated that it is predominantly distributed in the cytosol (Appendix A, lane 1). Further, to address whether this spatial expression of the circFLNB transcript is associated with a flux into the translating ribosomes [28], we isolated a ribosome-nascent chain complex (RNC) and monitored the association of circFLNB as an indicator of translation efficiency. Real-time PCR analysis consequently revealed a negligible amount of circFLNB in the RNC-mRNA fraction. This minimal association with the translation machinery likely implies low translatability of circFLNB (Appendix A). As a translating gene control, ACTIN expression was abundantly detected in the RNC-mRNA fraction. Taken together, our results in this series of experiments further resolved the qualitative and spatial characteristics of the OSCC-associated circFLNB.

### 3.4. Knockdown of circFLNB Compromised Tumor Attributes In Vitro and In Vivo

As the above results established the physical attributes of circFLNB, we next sought to ascertain the functional implications of its upregulation in OSCC. To this end, we designed shRNAs that uniquely targeted the back-splicing junction site of circFLNB and performed a series of loss-of-function analyses in oral cancer cell lines. Knockdown efficiency of the circFLNB-targeting specific shRNA was first confirmed by qPCR (Figure 4A, left panels). We subsequently discovered that shRNA-mediated downregulation of circFLNB in the SAS and SCC25 cells led to a reduced proliferation rate (Figure 4A, right panels) and a lower extent of colony formation (Figure 4B), compared to the control transfection group. This abrogated proliferation may be attributed to stalled cell cycle progression in circFLNB knockdown culture, as revealed by S-phase cell accumulation in the flow cytometry analysis (Appendix A). Moreover, silencing circFLNB expression enhanced apoptotic activation (i.e., cleaved PARP expression) and consequently diminished cell viability in cells treated with the chemotherapeutic agent doxorubicin (Figure 4C). Viewed together, these observations supported the notion that circFLNB acts as a positive regulator of cancer cell growth and survival, and implied that targeting circFLNB might sensitize cells to anti-cancer treatments. To further dissect the functional role of circFLNB, we constructed a circFLNB overexpression vector, which comprised of sequences corresponding to the circularized exons and adjoining complementary Alu sequences (Figure 5A, upper panel). Overexpression of circFLNB RNA in cancer cells was confirmed by PCR (Figure 5A, lower panel), and was found to confer a marginal effect on cell growth (Figure 5B and Appendix A). In contrast, cell survival in the presence of doxorubicin was improved in the circFLNB-overexpressing cells, which exhibited attenuated PARP protein cleavage and increased cell viability (Figure 5C). This observation strengthened support for a protective role of circFLNB against cytotoxic stress. Interestingly, expression of the parental FLNB transcripts remained unchanged in both circFLNB knockdown and overexpression conditions (Figure 5D), suggesting that the alternatively formed circRNA is regulated independently from the host FLNB gene.

### 3.5. Knockdown of circFLNB Reduced Cell Metastasis and Tumor Formation

Having established the importance of circFLNB in cancer cell growth, we next set out to examine the possible links to cancer metastasis. The migratory ability of cells was monitored using the wound healing assay, in which we observed that the knockdown of circFLNB expression markedly abrogated cell migration when compared to the control (Figure 6A). We then conducted Transwell migration and invasion assays to further verify this effect on the metastatic attributes of OSCC cells. Accordingly, silencing circFLNB expression effectively abolished the migratory and invasive properties of cancer cells (Figure 6B), which further linked circFLNB to the aggressive features of tumor cells.

Next, to assess tumor formation in vivo, stable clones of circFLNB knockdown and control SAS cells were established and then grafted subcutaneously into immunodeficient mice. Parameters such as tumor volume, size, and weight were recorded and analyzed over a six-week period as readouts for tumor growth in this in vivo model. Consistent with the previously observed effect on cell proliferation, we discovered that the tumor volume in the circFLNB knockdown group was significantly smaller than in the control group (Figure 6C, upper panel). Similar outcomes in tumor size and weight were also observed (Figure 6C, lower panel). Taken together, these results strongly evidenced a critical role for circFLNB in tumor development and progression.

### 3.6. Exploration of the circFLNB-Mediated Regulatory Network

Given that circRNAs have been associated with a general role in gene regulation, we inferred from our observations thus far that circFLNB may be involved in tumorigenesis-related gene expression. To test this hypothesis, we conducted RNA-seq experiments to profile transcriptome-wide alterations in cells subjected to circFLNB knockdown. Differential expression analysis first revealed that 309 genes were affected significantly in the knockdown cells relative to the control (|fold change| ≥ 1.5, *p*-value < 0.05, Appendix A). The overall distribution of the differentially altered genes, as depicted by a heatmap, showed distinct changes in the transcriptome landscape associated with circFLNB knockdown (Figure 7A). Pathway analysis of these differentially expressed genes using the Ingenuity Pathway Analysis (IPA) tool revealed significant downregulation of general oncogenic pathways, such as colorectal cancer metastasis signaling and glioma signaling (Figure 7B). This global alteration was consistent with the phenotypes observed in the circFLNB knockdown cells (Figure 4 and Figure 6), and further supported a positive role of circFLNB in oral cancer development and progression.

To further pinpoint potential downstream targets of circFLNB, we next compiled a co-expression gene set based on the transcriptome data from the 39 patients. This analysis uncovered 2489 genes that were positively correlated in expression with circFLNB (Figure 7C, left panel), as well as 3308 genes that were inversely correlated (Figure 7C, right panel). We then intersected this co-expression gene set from clinical samples with gene alterations in association with circFLNB knockdown, given that coordinated expression patterns revealed by the two datasets would imply targets directly regulated by circFLNB. We identified a common set of 27 genes from this integrative analysis (Figure 7C), of which 18 were positively regulated, and the remaining 9 were negatively correlated. Profiles of the specific gene sets revealed by clinical specimen sequencing and the circFLNB knockdown RNA-seq assay are presented as a hierarchical heatmap (Figure 7D,E). These observations further supported the idea that the distinct gene regulatory axes are under control by circFLNB and strengthen the role of circFLNB in promoting OSCC tumorigenic progression.

## 4. Discussion

Despite being neglected for decades, circRNAs have emerged as critical regulators of biological processes at both the transcriptional and post-transcriptional levels [8]. Even without protein-coding potential, these distinctly-structured RNAs are highly stable and have been found to be uniquely expressed in tumor specimens and liquid biopsies, which highlight their potential as viable biomarkers for disease screening or therapeutic targets [29]. The clinical focus of this study is OSCC, which is a commonly diagnosed cancer with regional prevalence and a persistently poor clinical outcome. The goal of the present work is thus to provide a novel perspective on the molecular mechanism underlying OSCC pathogenesis, focusing on circRNA alterations and their functional implications. Importantly, a systems approach based on RNA-seq data from clinical specimens was undertaken to globally demarcate circRNA–mRNA gene networks, which consequently shed light on the transcriptomic crosstalk that underlies oral cancer. The in silico analyses were complemented by functional characterization of a novel, pro-tumorigenic circRNA, named circFLNB, thus substantiating the tumorigenic role of the OSCC-associated circRNAome. Lending further support to this notion, a recent study has also discovered an OSCC-associated, splicing factor-derived circRNA, named circUHRF1, and demonstrated its involvement in tumorigenesis [30]. Interestingly, a circRNA (circFLNA) derived from another filamin family gene, filamin A, was reported to impact the migration of the laryngeal squamous cell carcinoma, which is related histologically and anatomically to oral cancer [31]. Collectively, these key findings underscore the critical role of transcriptomic alterations underlying oral cancer progression, and also augment their potential value regarding the detection, prevention, and treatment of this disease.

The mechanism underlying the biogenesis of circRNA, which essentially constitutes a back-splicing reaction, has been an interesting and as yet entirely unresolved topic of research [32]. In this capacity, RNA binding proteins (RBPs), in particular the alternative splicing factors, have been identified as binding the intronic regions flanking the circularized exons to facilitate their back-splicing and consequent formation of the circular structure [33]. The most notable example is the alternative splicing factor Quaking (QKI), which in the presence of the QKI motif could induce the preferential circularization of linear transcripts [34]. Interestingly, while no QKI motif was found in the vicinity of circFLNB splicing sites, we did identify binding sequences for ELAVL1/HuR, the so-called AU-rich sequence (AREs), in the flanking introns via sequence analysis. The RNA shuttling factor ELAVL1 has long been implicated in modulating various post-transcriptional attributes of mRNA transcripts, such as subcellular distribution and stability [35,36]. Its functional connection to the circRNA expression was also revealed in a recent study that characterized a circRNA derived from the gene encoding poly(A) tail-binding protein PABPN1 as the target of ELAVL1 [37]. In line with this notion, our preliminary results showed that ELAVL1 might promote the expression of circFLNB without altering the parental mRNA transcript (data not shown). Given the reportedly oncogenic potential of the ELAVL1 protein in various types of cancer [38,39,40], we speculated that the OSCC-associated upregulation of circFLNB could be partly attributed to this RBP.

A key finding of our study is the identification of a novel OSCC-associated circRNA and its downstream regulatory network. In this regard, the pro-tumorigenic function of circFLNB was first established by a series of in vitro and in vivo assays. Furthermore, given the regulatory role of circRNAs, we employed RNA-seq and in silico analyses (miRNA-target interactions) to map the downstream network governed by circFLNB. To this end, we superimposed the miRNA-target interactions as annotated in Figure 2D over the co-expression network, from which a potential ceRNA loop between circFLNB and miR-9-5p was uncovered (Appendix A). Interestingly, we discovered that miR-9-5p target sequences were shared between circFLNB and its positively-regulated downstream genes, SLC7A8 and ZNF618 (Appendix A). Given that circFLNB harbors two miR-9-5p target sequences (Appendix A), and also the reported down-regulation of miR-9-5p expression in oral cancers [41,42,43], our correlative evidence in this part suggest, among many possibilities, a presumably antagonistic interaction between these two non-coding RNAs. This is further characteristic of the “sponging” regulation widely documented in circRNA biology and would imply that miR-9 is negatively correlated with tumorigenesis. Indeed, a tumor-suppressive role has been extensively documented for miR-9. The miR-9 has been found to inhibit the progression of OSCC, glioblastoma, and ovarian serous carcinoma by, respectively, suppressing CDK4/6, FOXP2, and TLN1 [43,44,45]. In addition, by targeting NFκB1, miR-9 could enhance the sensitivity of tumor cells to ionizing radiation [46]. Moreover, by cross-referencing the co-expression network and miRNA interactions, our integrative analysis uncovered two downstream targets of circFLNB that are also competitively regulated by miR-9, SLC7A8, and ZNF618. The solute carrier SLC7A8 is known to promote glycolysis and chemoresistance in pancreatic cancer in an mTOR-dependent manner [47]. While functional role of ZNF618 in tumorigenesis is unknown, its expression has been found to be upregulated in glioma and some other cancers. Taken together, our work outlines a new circRNA–miRNA–mRNA regulatory axis with potential impact on tumorigenic progression in OSCC, and further strengthens the critical link between this malignancy and the circRNAome.

## Figures and Tables

**Figure 1 cells-09-01868-f001:**
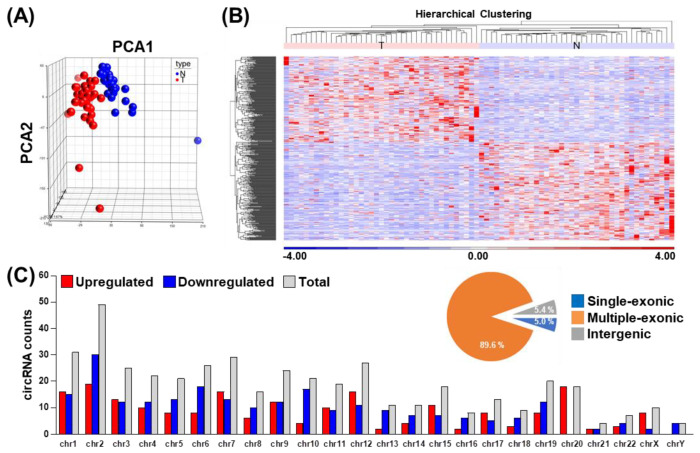
The circRNA transcriptome landscape in primary oral squamous cell carcinoma (OSCC) tumor tissues. Using the OSCC RNA-seq dataset, circRNA expression levels were determined based on the normalized read count values. (**A**) Principal component analysis (PCA) of normal vs. tumor tissues generated on the basis of identified circRNAomes. (**B**) Hierarchical clustering analysis of circRNAs differentially expressed in tumor vs. normal tissues (*n* = 443), which illustrates the distinction of differential expression profiles corresponding with disease state. (**C**) The distribution of differentially expressed circRNAs on the basis of chromosomal location. Sequence composition for circRNAs, including single-exonic, multiple exonic, and intergenic types are presented as circle plot in the upper right panel.

**Figure 2 cells-09-01868-f002:**
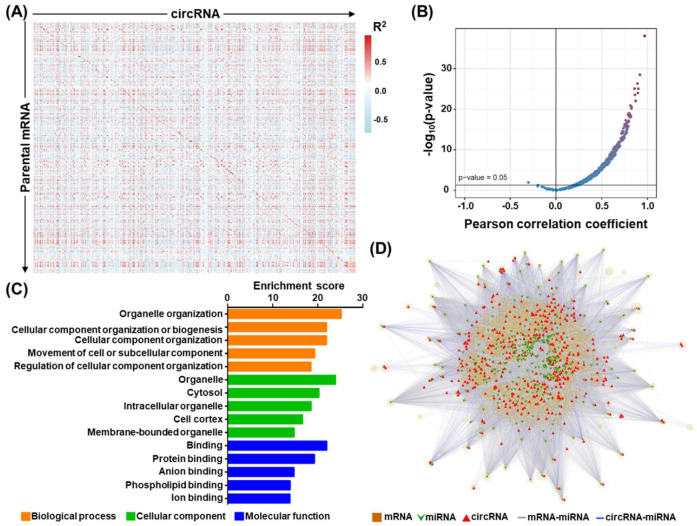
Transcriptomic networks in association with circRNAs in OSCC. (**A**) Gene co-expression analysis was performed between differentiated expressed circRNAs (horizontal axis) and their parental mRNA genes (vertical axis), based on the expression levels shown by OSCC RNA-seq data. The co-expression map is depicted as a heatmap, in which the correlation coefficients are represented by the colors shown by the scale bar in the right panel. (**B**) Extent of coordinated expression for circRNA and host mRNA pairs presented as a volcano/scatter plot, according to each pair’s correlation coefficient (x-axis) and significance (*p*-value; y-axis). (**C**) Gene ontology (GO) enrichment analysis of DEC-encoding host genes was performed. The x-axis indicates the significance of enrichment for the indicated term. (**D**) Construction of the circRNA–miRNA–mRNA regulatory network in OSCC. The miRNA interactions were predicted for each circRNA–mRNA pair with correlated expression in the OSCC specimens (padj < 0.05, cor > 0). A three-tier, ceRNA connection was established if an miRNA-targeted sequence was commonly found in the mRNA–circRNA pair (i.e., with a least two hits in circRNA). mRNA, squares; miRNA, inverted triangles; circRNA, triangles; miRNA–mRNA interaction, gray line; circRNA–miRNA interaction, blue line.

**Figure 3 cells-09-01868-f003:**
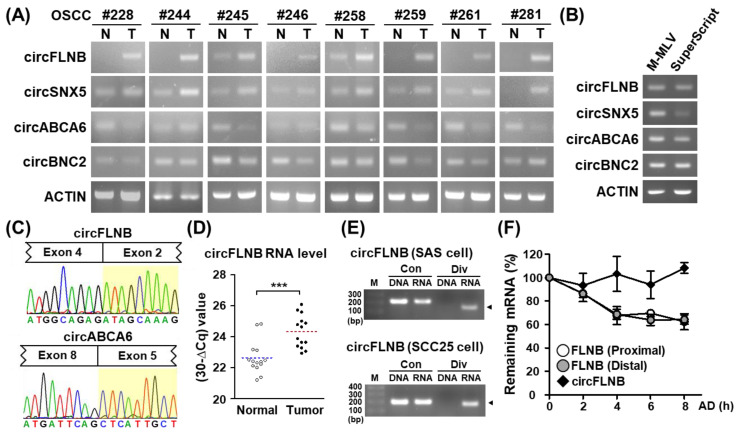
Validation and characterization of differentially expressed circRNAs in OSCC. (**A**) Expression of selected circRNAs in paired OSCC specimens was detected by end-point PCR. ACTIN gene expression was used as the loading control. (**B**) End-point PCR analyses of complementary DNA (cDNA) generated by M-MLV or SuperScript reverse transcriptases were performed to exclude the detection of artificial circRNA. ACTIN was used as the internal control. (**C**) Two exemplary circRNAs were PCR-amplified, and then subjected to Sanger sequencing. Histograms illustrate the junctional sequences flanking the back-splicing sites (with exons indicated above). (**D**) Upregulated circFLNB expression in tumor samples vs. adjacent normal tissues was confirmed by qRT-PCR analysis of an independent patient cohort. TBP gene expression was used as an internal control. For statistical analyses shown in this figure: *** *p* < 0.001. (**E**) PCR of genomic DNA (DNA) and reverse-transcribed cDNA (RNA) was performed with primers in the convergent (Con) and divergent (Div) orientations and analyzed by gel electrophoresis. (**F**) To monitor transcript stability, SAS cells were treated with actinomycin D (AD) for the indicated time lengths, then harvested for qRT-PCR-based gene expression analysis. RNA turnover rate was measured by normalization of RNA abundance relative to the initial time point and plotted for each indicated gene.

**Figure 4 cells-09-01868-f004:**
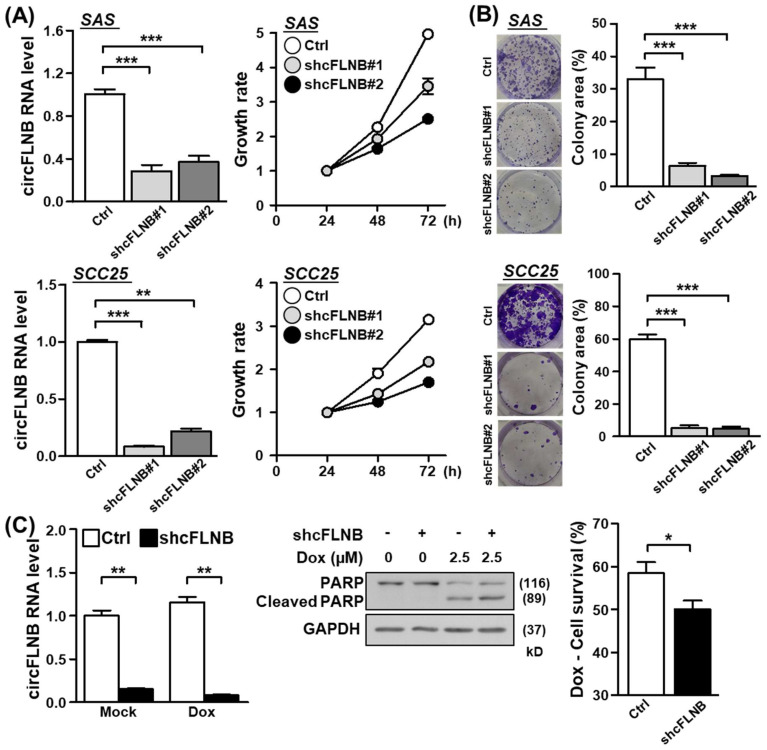
circFLNB is critical for maintaining oral cancer cell growth. The oral cancer lines SAS and SCC25 were subjected to circFLNB knockdown by means of lentiviral infection (shcFLNB#1 and shcFLNB#2), and knockdown efficacy was assessed by qPCR. Cell proliferation rate (**A**) and colony formation ability (**B**) of the knockdown cells were assessed by the MTT method and crystal violet staining, respectively. In (**B**), representative images (left) and quantified results based on colony area (right) are shown. (**C**) Control and circFLNB knockdown cells were treated with doxorubicin to induce apoptosis. PARP protein expression/cleavage (middle) and extent of cell survival (right) of the treated cells were analyzed by Western blot and MTT assays, respectively. GAPDH expression was used as the loading control. For statistical analyses shown in this figure: * *p* < 0.05; ** *p* < 0.01; *** *p* < 0.001.

**Figure 5 cells-09-01868-f005:**
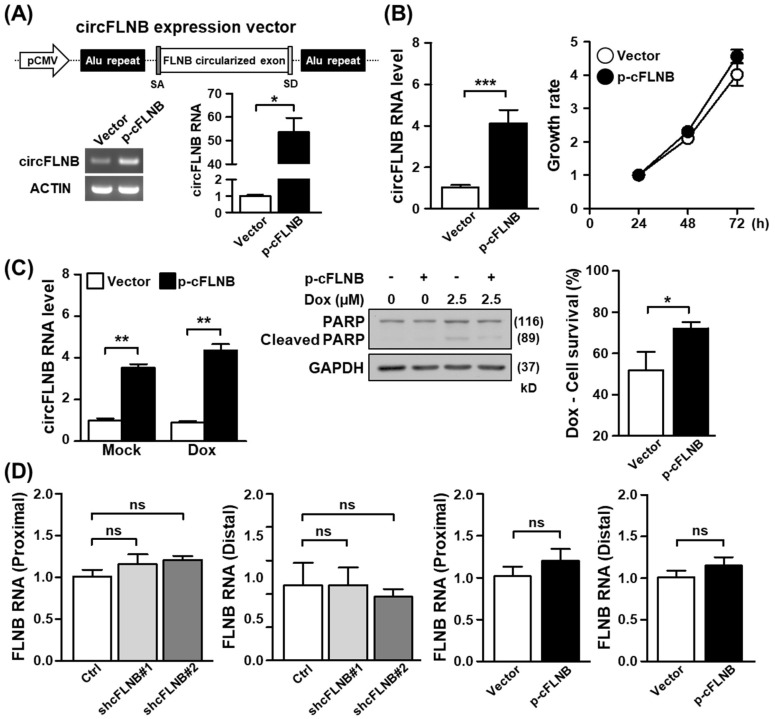
The circFLNB promotes cell survival under apoptotic stress. (**A**) Ectopic expression construct of circFLNB (p-cFLNB) was established by sub-cloning the circularized exons of circFLNB together with the flanking Alu repeats into a CMV-driven expression vector backbone. Overexpression efficacy in cells was monitored by end-point PCR and qPCR assays. (**B**) SAS cells were infected with control and p-cFLNB constructs, and subsequently analyzed for circFLNB expression and proliferation rate by qPCR and MTT methods, respectively. (**C**) Control and circFLNB overexpression cells were treated with 2.5 µM doxorubicin to induce cell death, and PARP protein expression/cleavage and cell survival were monitored as in Figure 4. (**D**) Expression levels of the parental FLNB gene in cells with circFLNB knockdown or overexpression was detected by qPCR assay, by using primers corresponding to FLNB transcript regions proximal or distal to the back-splicing junction. For statistical analyses shown in this figure: ^ns^
*p* > 0.05; * *p* < 0.05; ** *p* < 0.01; *** *p* < 0.001.

**Figure 6 cells-09-01868-f006:**
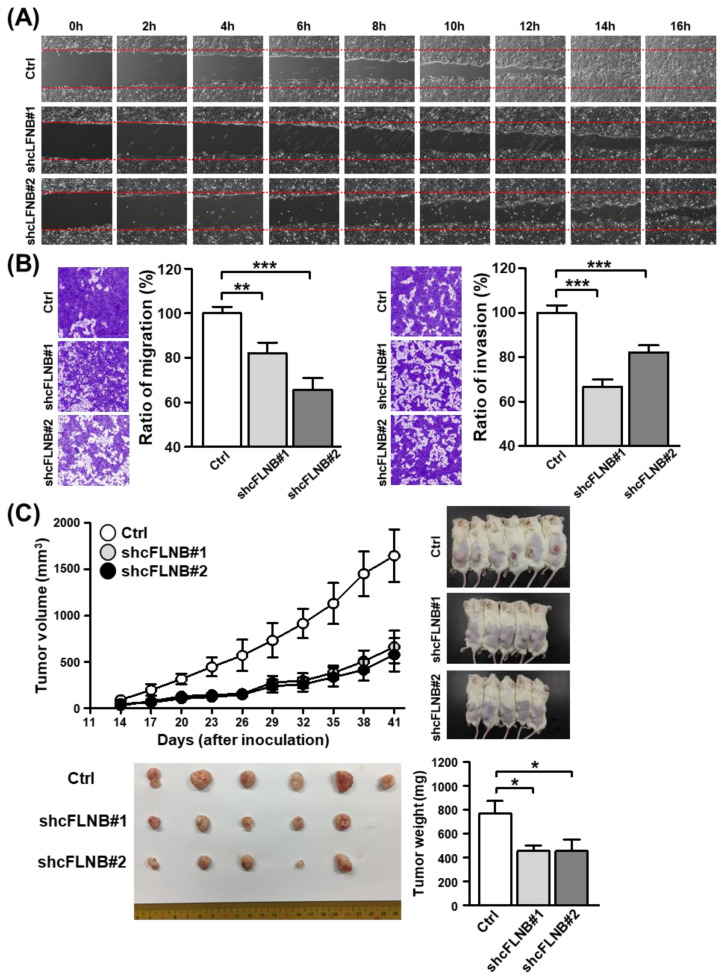
The circFLNB modulates migratory and invasive capability in OSCC. (**A**) The wound healing migration assay was performed to assess the migratory ability of SAS cells upon circFLNB knockdown. Photographs were taken at the indicated time points after cells were scratched, with representative images shown. (**B**) Transwell migration (left) and Matrigel invasion (right) assays were carried out after knockdown of circFLNB in SAS cells. The representative photomicrographs are shown. Average migratory area per field was quantified and expressed as a percentage relative to control cells (Ctrl). (**C**) The mouse xenograft experiment was performed by inoculating control and circFLNB knockdown SAS cells. Tumors formed at the indicated time points were dissected and measured for the volume (upper left). The right panel shows photographs of mice bearing tumors (top right) and the surgically removed tumors (bottom left). The bar graph depicts average weights (*n* = 10 or 11) of the tumor masses dissected from the indicated groups at sacrifice (day 41). For statistical analyses shown in this figure: * *p* < 0.05; ** *p* < 0.01; *** *p* < 0.001.

**Figure 7 cells-09-01868-f007:**
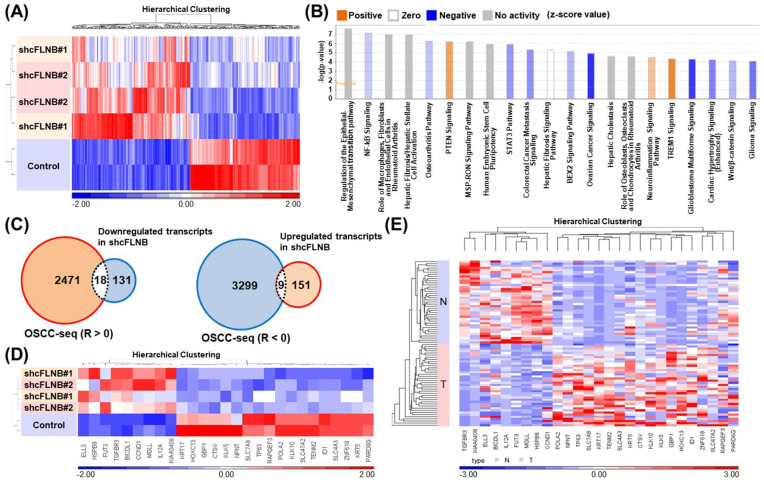
Transcriptome-wide exploration of circFLNB-mediated regulatory network. (**A**) Transcriptome-wide changes in circFLNB knockdown cells were profiled by RNA-seq. Overall profiles of the differentially expressed genes in the control and knockdown groups (as indicated) are represented by the heatmap. The expression values (represented by normalized read counts) are displayed in shades of red or blue (linear scale) relative to the means of all corresponding values within individual experimental groups. Clusters of genes (based on unsupervised hierarchical clustering) are indicated and denoted by their experimental types/conditions. (**B**) Differentially expressed genes (DEGs) were subjected to pathway analysis using the Ingenuity Pathway Analysis (IPA) tool. (**C**) Venn diagrams illustrate the degree of overlap between the circFLNB-correlated genes as shown by OSCC-specimen RNA-seq (OSCC-seq) and those differentiated expressed in the circFLNB knockdown experiments. To identify potential downstream targets of circFLNB, positively-correlated genes (R > 0) were intersected with downregulated DEGs in circFLNB knockdown (left; *n* = 18), while inversely correlated genes (R < 0) were intersected with upregulated DEGs (right; *n* = 9). (**D**, **E**) The distribution of the expression profiles of the 27 intersected genes uncovered in (**C**) are shown as heatmaps in the circFLNB knockdown experiments (**D**) and the OSCC samples from the 39 patients (**E**).

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
