# Peer review of "circRNAome Profiling in Oral Carcinoma Unveils a Novel circFLNB that Mediates Tumour Growth-Regulating Transcriptional Response"

_cells, 2020, doi:10.3390/cells9081868_

Round 1

Reviewer 1 Report

In the manuscript by Yi-Tung Chen, et.al. »circRNAome profiling unveils a novel oncogenic circFLNB circuit in oral cancer authors« describe translational medicine approaches for dysregulated circRNAs identification with tumorigenic progression potential in oral squamous cell carcinoma. Using RNA-seq data, they profiled the circRNAomes in oral squamous cell carcinoma. Using extensive bioinformatics analyses tumor-associated circRNA – circFLNB, circFLNB-targeted gene network and the tumor suppressive miR-9-5p endogenous RNA target were further identified. In silico, analysis and results are well supported by cell culture and mouse xenograft experiment. This paper is very well structured and written and conclusions are supported by the data. There is an abundance of very well conducted experimental work.

I have only two minors:

  • Fig 2D is not very clear. Authors should rethink perhaps, if there is any way to improve its appearance.
  • There is a similar paper published in 2019 (Jian-Xing Wang et.al., Cancer Cell Int Upregulation of circFLNA Contributes to Laryngeal Squamous Cell Carcinoma Migration by circFLNA-miR-486-3p-FLNA Axis Cancer Cell Int. 2019 Jul 29;19:196. doi: 10.1186/s12935-019-0924-9. eCollection 2019.) describing circFLNA contributing to Laryngeal Squamous Cell Carcinoma. Authors should rethink perhaps to include this paper in the Discussion section.

Author Response

Point-by-point response to referees comments (cells-827113)

(>>: our responses)

(Modifications and the inclusion of new data have been marked in red in the revised manuscript and summarized in List of amendments to manuscript, pages 1-2)

Reviewer #1

In the manuscript by Yi-Tung Chen, et.al. »circRNAome profiling unveils a novel oncogenic circFLNB circuit in oral cancer authors« describe translational medicine approaches for dysregulated circRNAs identification with tumorigenic progression potential in oral squamous cell carcinoma. Using RNA-seq data, they profiled the circRNAomes in oral squamous cell carcinoma. Using extensive bioinformatics analyses tumor-associated circRNA – circFLNB, circFLNB-targeted gene network and the tumor suppressive miR-9-5p endogenous RNA target were further identified. In silico, analysis and results are well supported by cell culture and mouse xenograft experiment. This paper is very well structured and written and conclusions are supported by the data. There is an abundance of very well conducted experimental work.

>>Thank you for the accurate summary and constructive comments to this study. Our responses to your specific concerns are as follows. 

I have only two minors:

1. Fig 2D is not very clear. Authors should rethink perhaps, if there is any way to improve its appearance.

>>We apologize for the inadequate resolution of the circRNA-miRNA-mRNA network shown in Fig. 2D, which was intended to illustrate the extensive regulatory connections among these molecules in OSCC. Because of the complex nature of the three-tier regulatory network, which is formed by 473,294 sub-networks, this image posed a challenge in terms of image preparation. To further improve its presentation, we have enlarged the markings and lines in the figure to make them significantly more distinct, and also augmented the overall resolution for the image output.

2. There is a similar paper published in 2019 (Jian-Xing Wang et.al., Cancer Cell Int Upregulation of circFLNA Contributes to Laryngeal Squamous Cell Carcinoma Migration by circFLNA-miR-486-3p-FLNA Axis Cancer Cell Int. 2019 Jul 29;19:196. doi: 10.1186/s12935-019-0924-9. eCollection 2019.) describing circFLNA contributing to Laryngeal Squamous Cell Carcinoma. Authors should rethink perhaps to include this paper in the Discussion section.

Reviewer #1 (Significance):

>>We thank the reviewer for the important information on a related report. We have followed his/her suggestion by incorporating this paper and a brief statement in the Discussion section.

Reviewer 2 Report

-The authors should explain why they selected exactly the circFLNB;

-In addition to the end-point PCR, real-time PCR should be used for more precise detection of circRNAs ( see fig. 3A)

-With reference to fig. 3, a Sanger sequence should be performed for all the 4 circRNAs studied.

-In fig. 4, cell cycle analysis should be performed in order to strengthen the results of the cell proliferation rate.

-The sponging mechanism of circFLNB on miR-9-5p should be investigated in more details with appropriate experiments

Author Response

Point-by-point response to referees comments (cells-827113)

(>>: our responses)

(Modifications and the inclusion of new data have been marked in red in the revised manuscript and summarized in List of amendments to manuscript, pages 1-2)

Reviewer #2

1. The authors should explain why they selected exactly the circFLNB;

>>We thank the reviewer for pointing this out. We have now added statement in line 295 to describe our rationale behind target selection: “we selected circFLNB for further functional characterization based on its prominent and consistent upregulation in tumor specimens in comparison with other candidates (Figure 3A)”.

2. In addition to the end-point PCR, real-time PCR should be used for more precise detection of circRNAs (see fig. 3A)

>>We performed end-point PCR analyses in Fig. 3A to qualitatively and semi-quantitatively verify the target circRNAs expression in tumor specimens. Due to the limitation in sample availability, we were not able to carry out additional real-time PCR for all the candidate circRNAs. Instead, for the selected circFLNB, we elected to further confirm its differential expression in the OSCC specimens from another cohort by using the real-time PCR assay (Figure 3D).

3. With reference to fig. 3, a Sanger sequence should be performed for all the 4 circRNAs studied.

>>We thank the reviewer for the kind suggestion. The Sanger sequencing results for circSNX5 and circBNC2 products were available at the time of submission but was not included. We have now added these chromatograms as new the Figure S1B.

4. In fig. 4, cell cycle analysis should be performed in order to strengthen the results of the cell proliferation rate.

>>To address this concern, we have conducted flow cytometry-based cell cycle profiling and consequently demonstrated a stalled cell cycle (S-phase) progression in the circFLNB knockdown cells. These results complemented our data of the reduced proliferation in circFLNB knockdown culture. This new cell cycle analysis is now shown as the new Figure S4.  

5. The sponging mechanism of circFLNB on miR-9-5p should be investigated in more details with appropriate experiments.

>>We thank the reviewer for the kind suggestion. The main goal of this manuscript is to provide a global view of the circRNA landscape in OSCC and use a candidate circRNA as an example to show the functional implications in OSCC progression. While we agree with the reviewer that demonstration of the sponging effect of circFLNB on miR-9-5p would be a nice addition to the study, it actually does not fall within the scope of the present study. In addition, a detailed mechanistic analysis of the circFLNB on miR-9-5p (such as the competitive 3’ UTR reporter assay) likely will require extended efforts and time to complete. We believe the substantial analytic and experimental information provided by our work has already served as sufficient evidence for the functional link of circRNAome to oral cancer.

Round 2

Reviewer 2 Report

The revised manuscript has been improved but it still lacks any mechanistic details regarding the sponge effect of the identified circularRNA. Additional experimental evidence needs to be provide before further evaluation in Cells.

Author Response

Point-by-point response to editor comment (cells-827113)

(>>: our responses)

(Modifications and the inclusion of new data have been marked in red in the revised manuscript and summarized in List of amendments to manuscript page 2-4)

The revised manuscript has been improved but it still lacks any mechanistic details regarding the sponge effect of the identified circularRNA. Additional experimental evidence needs to be provided before further evaluation in Cells.

>>We thank the reviewer for this helpful suggestion. We agree with the reviewer that the sponging effect shown in Figure 7 might be somewhat correlative and not sufficiently supported by experiments. To provide a more accurate account of our results, we have now moved all the descriptions related to this in silico finding to the Discussion section and made the corresponding modifications in the text. In particular, Figure 7E is now moved to the Discussion and shown as the new Figure S6. In addition, we moderated our conclusion by removing statements about circRNA sponging function from the Abstract and Introduction sections (lines 35 and 93, respectively). We also removed ceRNA function from the title of Figure 7 (line 441), which is now “Transcriptome-wide exploration of circFLNB-mediated regulatory network”.

List of amendments to manuscript (MS ID#: cells-827113)

Referees’ comments have been addressed in detail in the “Point-by-point response to referees comments”. Modifications and the inclusion of new data have been marked in red in the revised manuscript and summarized by the following points:

Author attributes:

P1 (line 7): We changed the order of the first two affiliations: 1Department of Biomedical Sciences, College of Medicine, Chang Gung University, Taoyuan, Taiwan Research Center for Emerging Viral Infections, Chang Gung University, Taoyuan, Taiwan

2Research Center for Emerging Viral Infections, Chang Gung University, Taoyuan, Taiwan Department of Biomedical Sciences, College of Medicine, Chang Gung University, Taoyuan, Taiwan

Abstract:

P1 (line 35): Descriptions for the sponging function of circFLNB were removed, and the main text was modified accordingly.

Introduction:

P2 (line 93): Descriptions for the sponging function of circFLNB were removed, and the main text was modified accordingly.

Materials and Methods:

P4 (line 183): “Ethics approval and consent to participate” is now included in this section.

Results:

P7 (line 260): We corrected the legend of Fig. 2D, which describes the criterial for selecting circRNA-mRNA pairs with correlated expression in the OSCC specimens (padj < 0.05, cor > 0).

P7-8 (lines 273 and 280): Expanded sequencing chromatogram for circFLNB junction is now added as the new Figure S1B, and Sanger sequencing analyses of circSNX5 and circBNC2 are included as the new Figure S1C. Because of this addition, the figures in the Figure S1 were rearranged accordingly. We also modified description for PCR amplicons sequencing: “PCR amplicons were ligated into the pHE cloning vector for complete Sanger sequencing…”.

P8 (line 303): We included the descriptions “based on its prominent and consistent upregulation in tumor specimens in comparison with other candidates (Figure 3A)” to state the rationale of selecting circFLNB for further functional characterization.

P9 (line 342): Results of the cell cycle analysis of circFLNB knockdown cells are included and presented the new Figure S4.

P9 (line 354): The original Figure S4 was rearranged to Figure S5 (due to this addition of new Figure S4).

P10 (line 359): The original Figure S5 was moved to the main text as the new Figure 5D.

P13 (line 433): Descriptions for the sponging function of circFLNB were moved to the Discussion section, and the main text was modified accordingly. The corresponding Figure 7E is now shown as Figure S6.

P14 (line 441 and 456): Because of the above amendment, the title and legends of Figure 7 were modified accordingly.

P15 (line 478): We incorporated a new reference on the circFLNA work and made a brief statement in the Discussion section.

P15 (lines 504 to 511): Descriptions for the sponging function of circFLNB were moved from the Results to the Discussion section, and the main text was modified accordingly. The corresponding Figure 7E is now shown as Figure S6.

P16 (line 527): The title of the new Figure S4 and Figure S6 (originally Figure 7E) are listed.

References:

P18. The publication entitled “Upregulation of circFLNA Contributes to Laryngeal Squamous Cell Carcinoma Migration by circFLNA-miR-486-3p-FLNA Axis” is now included as reference 31.

Supplemental Information

Materials and Methods

We modified the descriptions for the cDNA synthesis step in the RNC-mRNA fractionation experiment: “Collected RNA was reverse transcribed into cDNA sample by random hexamer primers…”. Method for cell cycle analysis is now included in this section.

Figures and Legends:

New data, with the corresponding legends, were incorporated in the revised version: Figures S1B, S1C, S4, and S6. Because of these changes, numbering in Figures S1 and S5 was modified.